# Research on the Modulation and Control Strategy for a Novel Single-Phase Current Source Inverter

**Yi Zhang [1,\*], Tao Yang [2,3] and Yiru Miao [3]**

1   School of Vehicle Engineering, Chongqing University of Technology, Chongqing 400054, China
2   School of Rail Transit and Aviation Service, Chongqing Industry Polytechnic College,
    Chongqing 401120, China; justmuch@163.com
3   State Key Laboratory of Power Transmission Equipment & System Security and New Technology,
    Chongqing University, Chongqing 400044, China; miaoyiru@cqu.edu.cn
\*   Correspondence: zagyi81@cqut.edu.cn

**Abstract:** Compared to the voltage source inverter, the current source inverter (CSI) can boost voltage and improve filtering performance. However, the DC side of CSI is not a real current source, and the DC input current comprises a DC power supply and an inductor. In the switching process, the DC-link inductor is charged or discharged and is in an uncontrollable state. This paper proposes a novel CSI topology containing five switching tubes and a modulation strategy based on the hysteresis control strategy of the DC-link current. Due to the conduction and switching loss being positive to the DC-link current, the calculation method for the least reference value of the DC-link current is derived to meet power requirements. By constructing a virtual axis, we then present the control strategy of the output voltage in a two-phase rotating reference frame. Finally, we carry out the simulation and experiment are to validate the proposed topology, modulation, and control strategy.

**Keywords:** current source inverter; DC-link current; modulation; control strategy

## 1. Introduction

Inverters can be categorized as voltage source inverters (VSI) [1] and current source inverters (CSI) [2] according to the characteristics of input power supplies. Some unique advantages of CSIs have been discovered [3], such as their boost capacity, inherent short-circuit protection capacity, and AC filtering structure. CSI, therefore, has potential applications [4] in the solar photovoltaic industry, wind energy power generation, motor drive systems, and HVDC transmission systems.

In recent years, various control and modulation strategies for CSI have been proposed [5], such as decoupling methods, low common voltage modulation, and digital vector control strategy. However, the DC-link current is considered a constant value in the above studies in which the charging or discharging process of the DC-link inductor has been ignored, leading to the DC-link current being discontinuous or continuously increasing.

Unlike VSI, the DC-link current of CSI is formed by the DC input voltage DC-link inductor, and constant DC-link current output is a necessary condition for high performance operation of CSI [6]. In [7–11], several novel CSI topologies are proposed to achieve control of the DC-link current. In [7], a three-phase current source rectifier (CSR) is introduced to adjust the DC-link current, but this topology is very complex and only suitable for AC-DC-AC applications. A buck converter is added to the DC side to regulate the DC-link current in [8]. In [6], the buck converter is replaced with a bi-directional converter that can control the current, and the buck operation can also be realized. However, the switching and conduction losses increase as the converter is introduced. In [9], a buck-boost CSI is proposed, which has the advantages of a simple structure and large voltage output range, but the control strategy is difficult to implement. The current-fed quasi-Z-source inverter is

proposed in [10]. However, with the addition of many passive components, both efficiency and power density are greatly reduced.

In [11], an additional switching tube and diode are connected in parallel with the DC-side inductor of the three-phase CSI. The DC-side inductor can self-continuate when the switch is turned on. During the zero vector period, the continuous current mode is used to replace the traditional magnetizing mode, which can prevent the increase of the DC-link current. On this basis, a novel single-phase five-switch CSI topology is proposed in this paper. The modulation and control strategies are studied to obtain the constant DC-link current and high-quality AC voltage output. Finally, the simulation and experimental verification are carried out.

## 2. Topology and Operating Modes of the Proposed Single-Phase CSI

### 2.1. Analysis of the Proposed Single-Phase CSI's Operating Modes

The topology of the proposed single-phase CSI is shown in Figure 1. The DC side is composed of a DC voltage source and DC-link inductance $L_{dc}$. The inverter part is composed of switching tubes ($S_1$, $S_2$, $S_3$, $S_4$) and diodes ($D_1$, $D_2$, $D_3$, $D_4$). The AC side contains the filter capacitor $C$ and the resistive load $R$. The switching tube $S_0$ and diode $D_0$ are parallel with $L_{dc}$.

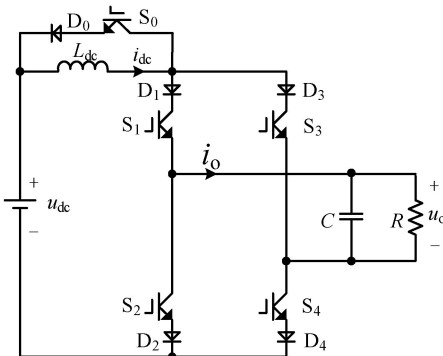

**Figure 1.** The topology of the proposed single-phase CSI.

There are four operating modes of the proposed single-phase CSI. The corresponding switching states are shown in Table 1.

**Table 1.** The operating modes, switching state, and switching function.

| Operating Modes | Switching State | Switching Function $p$ and $q$ |
|---|---|---|
| magnetizing mode | $S_1$ and $S_2$ ON, $S_0$, $S_3$, and $S_4$ OFF | $p = 0$, $q = 1$ |
| energy-supplying mode I | $S_1$ and $S_4$ ON, $S_0$, $S_2$, and $S_3$ OFF | $p = 1$, $q = 0$ |
| energy-supplying mode II | $S_2$ and $S_3$ ON, $S_0$, $S_1$, and $S_4$ OFF | $p = -1$, $q = 0$ |
| freewheeling model | $S_0$ ON, $S_1$, $S_2$, $S_3$, and $S_4$ OFF | $p = 0$, $q = 0$ |

(1) Magnetizing mode: $S_1$ and $S_2$ are turned on, and the equivalent circuit under magnetizing mode is shown in Figure 2a. At this moment, $i_o$ is equal to 0 A, the $u_{dc}$ is charging to $L_{dc}$, and $C$ provides energy for the load separately.

(2) Energy-supplying mode I: $S_1$ and $S_4$ are turned on, and the equivalent circuit under this operating mode is shown in Figure 2b. At this moment, $i_o$ is equal to $i_{dc}$, and the $u_{dc}$ and $L_{dc}$ provide energy to the AC load together.

(3) Energy-supplying mode II: $S_2$ and $S_3$ are turned on, and the equivalent circuit under this operating mode is shown in Figure 2c. Different from energy-supplying mode I, the polarity of the output current is negative; $i_o$ is equal to $-i_{dc}$.

(4) Freewheeling model: Only $S_0$ is turned on, and the equivalent circuit under the freewheeling model is shown in Figure 2d. The $i_{dc}$ freewheels through $S_0$, and $C$ provides energy for the load separately.

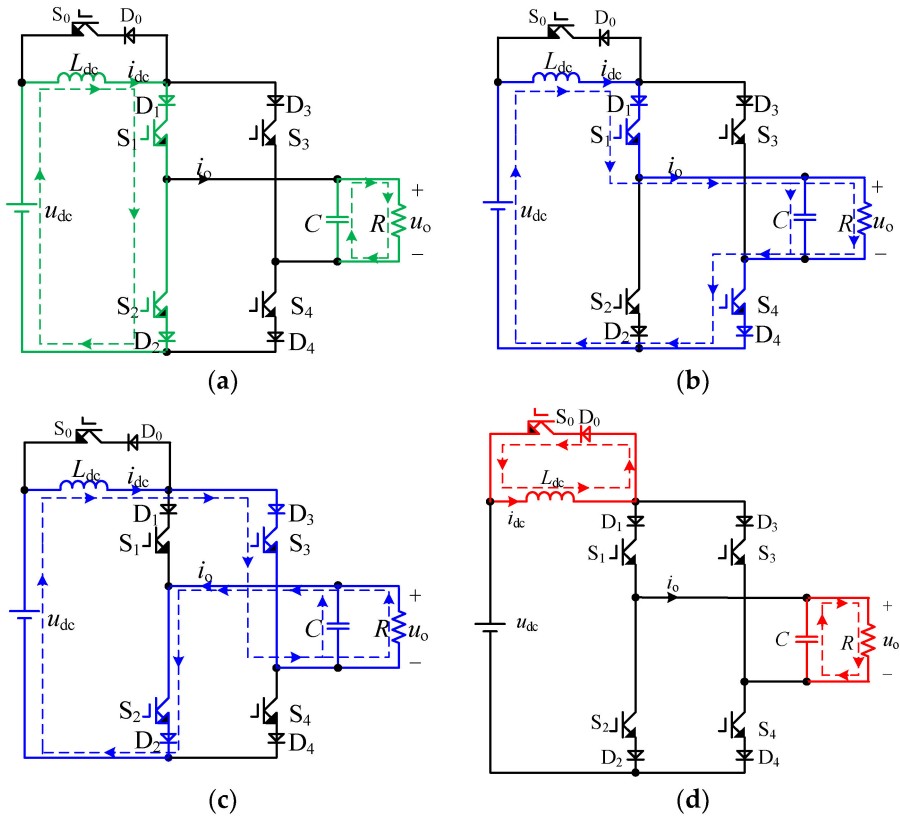

**Figure 2.** The operating modes of the proposed single-phase CSI: (**a**) magnetizing mode; (**b**) energy-supplying mode I; (**c**) energy-supplying mode II; (**d**) freewheeling model.

Two switching functions ($p$ and $q$) are defined and shown in Table 1. The $p$ is used to indicate whether the CSI is operating in energy-supplying mode, and the math relationship between $i_{dc}$ and $i_o$ can be expressed as

$$i_o = i_{dc}p \tag{1}$$

The state equation of $u_o$ follows:

$$C\frac{du_o}{dt} = pi_{dc} - \frac{u_o}{R} \tag{2}$$

$q$ is used to indicate whether the CSI is operating in magnetizing mode. The state equation of $i_o$ follows:

$$L_{dc}\frac{di_{dc}}{dt} = qu_{dc} - pu_o \tag{3}$$

Since the conduction current of all switching tubes is $i_{dc}$, the current stresses of all switching tubes are equal to the DC-link current. The voltage stresses can be derived according to Figure 2, and the voltage stresses of all switching tubes under different operating modes are summarized in Table 2.

**Table 2.** Voltage stresses of all switching tubes under different operating modes.

| Operating Mode | Withstand Voltage $S_0$ | Withstand Voltage $S_1$ | Withstand Voltage $S_2$ | Withstand Voltage $S_3$ | Withstand Voltage $S_4$ |
|---|---|---|---|---|---|
| Magnetizing mode | $u_{dc}$ | 0 | 0 | $-u_o$ | $u_o$ |
| Energy-supplying mode | $u_{dc} - u_o$ | 0 | $-u_o$ | $-u_o$ | 0 |
| Freewheeling model | 0 | $u_o$ | $-u_{dc} - u_o$ | 0 | $-u_{dc}$ |

### 2.2. Comparison of Different CSI Topologies

The recent literature presents many CSI structures comprising distinct tradeoffs among different topologies, as shown in Figure 3. According to [12,13], a cost function (*CF*) is established in (4) to carry out a fair comparison of different CSI topologies:

$$CF = N_V(N_{\text{US}} + 2N_{\text{BS}} + N_D + N_C + N_{\text{drv}} + N_T)/N_l \tag{4}$$

where $N_V$ is the number of DC voltages, $N_{\text{US}}$ is the number of unidirectional switches, $N_{\text{BS}}$ is the number of bidirectional switches, $N_D$ is the number of diodes, $N_C$ is the number of capacitors, $N_{\text{drv}}$ is the number of individual drivers, $N_T$ is the number of transformers, and $N_l$ is the number of output current levels.

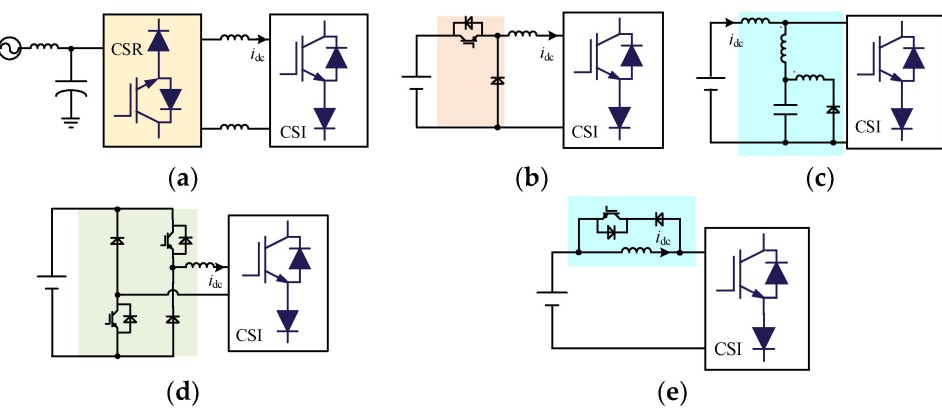

(a)  (b)  (c)

(d)  (e)

**Figure 3.** The topologies of different CSI: (**a**) current source rectifier; (**b**) buck converter; (**c**) quasi-Z source network; (**d**) bidirectional DC chopper; (**e**) proposed single-phase CSI.

The comparison result is presented in Table 3. Although the quasi-Z source network receives the lowest *CF*, it contains numerous passive components, and its control system is too complex. The proposed single-phase CSI and the buck converter have the same CF. However, the buck converter contains more switches than the proposed single-phase CSI. The bidirectional DC chopper's *CF* is slightly higher than the proposed one because of the additional two diodes and switching tubes. The current source rectifier rates the worst *CF* because it includes too many components.

**Table 3.** Comparison of different CSI topologies.

| Parameter | Current Source Rectifier | Buck Converter | Quasi-Z Source Network | Bidirectional DC Chopper | Proposed Single-Phase CSI |
|---|---|---|---|---|---|
| $N_V$ | 1 | 1 | 1 | 1 | 1 |
| $N_{\text{US}}$ | 0 | 0 | 0 | 0 | 0 |
| $N_{\text{BS}}$ | 8 | 5 | 4 | 6 | 5 |
| $N_D$ | 8 | 5 | 5 | 6 | 5 |
| $N_C$ | 2 | 1 | 2 | 1 | 1 |
| $N_{\text{drv}}$ | 8 | 5 | 4 | 6 | 5 |
| $N_T$ | 0 | 0 | 1 | 0 | 0 |
| $N_l$ | 3 | 3 | 3 | 3 | 3 |
| CF | 11.33 | 7 | 6.67 | 8.33 | 7 |
| Input H Bridge | Yes | No | No | Yes | No |
| Modulation | PWM | PWM | PWM | PWM | PWM |

## 3. Modulation Strategy Based on DC-Link Current Control

Unlike VSI, switching signals cannot be directly generated by comparing modulated signals with carrier waves [14]. In [15], a logic conversion circuit is adopted to convert the switching signals, which will delay them. In this section, a novel modulation strategy is

presented which can be implemented by the DSP without a logic conversion circuit. Three logic signals ($p_1$, $p_2$, and $p_3$) are set, where $p_1$ represents the comparison result between the modulation wave and carrier wave, as shown in Figure 4, and where $d$ is the duty cycle and is defined as

$$d = \frac{|i_o|}{i_{dc}} \qquad (5)$$

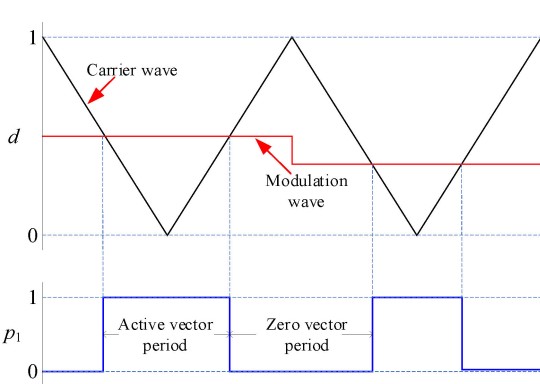

**Figure 4.** The diagram of $p_1$, carrier wave, and modulation wave.

$p_1 = 1$ represents the active vector period, and the CSI operates at energy-supplying mode I or II, which is determined by the polarity of $i_o$. $p_2$ represents the polarity of $i_o$. If $i_o > 0$, $p_2 = 1$; otherwise, $p_2 = 0$.

$p_1 = 0$ represents the zero vector period, and the CSI operates on either the magnetizing mode or freewheeling model, which is determined by $i_{dc}$. If $i_{dc} >$ the reference value $i_{dc}^*$, the freewheeling model will be selected to prevent $i_{dc}$ from increasing in the zero vector period. Otherwise, the magnetizing mode will be implemented to charge $L_{dc}$ in the zero vector period. $p_3$ is defined to represent the math relationship between $i_{dc}$ and $i_{dc}^*$. If $i_{dc} > i_{dc}^*$, $p_3 = 1$; else, $p_3 = 0$. The changing process of $i_{dc}$ and $p_3$ is described in Figure 5. $i_{dc}$ can be adjusted according to $p_3$. This approach belongs to a hysteresis control method [16].

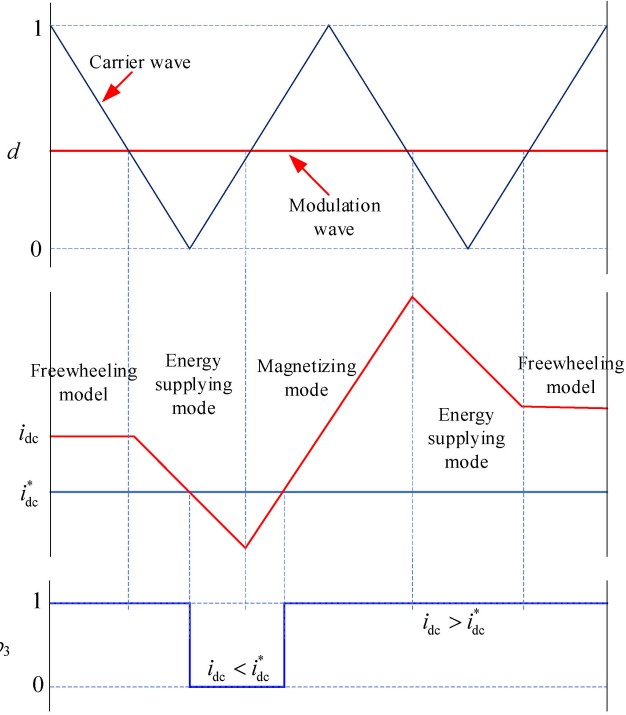

**Figure 5.** The changing process of $i_{dc}$ and $p_3$.

Based on the above settings and analysis, the production logic of the five switching signals is shown in Figure 6. The specific logic expression for each switching is described as follows:

$$\begin{cases} p_{S1} = \overline{p}_1 \& \overline{p}_3 | p_2 \\ p_{S2} = \overline{p}_1 \& \overline{p}_3 | \overline{p}_2 \\ p_{S3} = p_1 \& \overline{p}_2 \\ p_{S4} = p_1 \& p_2 \\ p_{S0} = \overline{p}_2 \& p_3 \end{cases} \tag{6}$$

where $p_{S1}$, $p_{S2}$, $p_{S3}$, $p_{S4}$, and $p_{S5}$ represent the driving logic of switch $S_1$, $S_2$, $S_3$, $S_4$, and $S_5$, respectively.

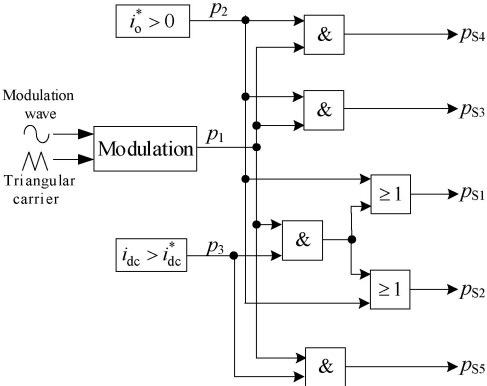

**Figure 6.** Logical conversion for the proposed single-phase CSI.

## 4. Calculation for the Optimal Reference of DC-Link Current

In Section 3, a DC-link current hysteresis control is introduced into the modulation scheme. However, the reference of the DC-link current must be determined for the following reasons [17]. If $i_{dc}^*$ is set too small, the DC-link current is unable to provide sufficient power to the AC load. If $i_{dc}^*$ is set too great, the conduction loss, switching loss, and harmonic distortion will increase. Thus, the DC-link current should be reduced to satisfy the current requirements. The calculation method for the optimal reference of the DC-link current is derived in this section.

It is assumed that $u_o$ remains in a constant state in a switching period. Under energy-supplying mode, the reduction $\Delta i_{dc\_down}$ of $i_{dc}$ can be expressed as follows:

$$\Delta i_{dc\_down} = \frac{T_s}{L_{dc}} d(t) [u_o(t) - u_{dc}] \tag{7}$$

The increment of $\Delta i_{dc\_up}$ under magnetizing mode can be expressed as

$$\Delta i_{dc\_up} = \frac{u_{dc} T_s}{L_{dc}} [1 - d(t)] \tag{8}$$

According to (7) and (8), the total reduction $\Delta i_{dc}$ of $i_{dc}$ in a switching period can be expressed as

$$\Delta i_{dc} = \Delta i_{dc\_down} - \Delta i_{dc\_up} = \frac{T_s}{L_{dc}} [u_o(t)d(t) - u_{dc}] \tag{9}$$

By substituting (5) with (9), $\Delta i_{dc}$ can be rewritten as

$$\Delta i_{dc} = \frac{T_s}{L_{dc}} \left[ \frac{u_o(t) i_o(t)}{i_{dc}(t)} - u_{dc} \right] \tag{10}$$

Ignoring the harmonic component and initial phase, $u_o$ is expressed as

$$u_o(t) = U \sin \omega t \tag{11}$$

where $U$ represents the fundamental amplitude of $u_o$, and $\omega$ represents the fundamental frequency. $i_o$ is expressed as

$$i_o(t) = I \sin(\omega t + \theta) \tag{12}$$

where $I$ represents the amplitude of $i_o$, and $\theta$ represents the initial phase of $i_o$. Due to the load of the inverter consisting of the resistance and the capacitance paralleling with the load, the following relations are satisfied:

$$\begin{cases} I = U\sqrt{1 + (\omega CR)^2}/R \\ \theta = \arctan(\omega CR) \end{cases} \tag{13}$$

Substituting (11) and (12) with (10), the following can be obtained:

$$\Delta i_{dc} = \frac{T_s}{L}\left[\frac{UI\sin(\omega t)\sin(\omega t + \theta)}{i_{dc}(t)} - u_{dc}\right] \tag{14}$$

Formula (14) is simplified as follows:

$$\Delta i_{dc} = \frac{T_s}{2L}\left\{\frac{UI[\cos\theta - \cos(2\omega t + \theta)]}{i_{dc}(t)} - 2u_{dc}\right\} \tag{15}$$

The maximum reduction $\Delta i_{dcmax}$ of $i_{dc}$ in a switching period is shown as follows:

$$\Delta i_{dcmax} = \frac{T_s}{2L}\left[\frac{UI(\cos\theta + 1)}{i_{dc}(t)} - 2u_{dc}\right] \tag{16}$$

If $\Delta i_{dcmax} < 0$, $i_{dc}$ can continue increasing in any switching period, which consists of magnetizing mode and energy-supplying mode. Thus, $i_{dc}$ needs to be satisfied as follows:

$$i_{dc} > \frac{UI(\cos\theta + 1)}{2u_{dc}} \tag{17}$$

Substituting (13) with (17), and replacing $i_{dc}$ with $i_{dc}^*$, Formula (17) can be rewritten as follows:

$$i_{dc}^* > \frac{U^2\left[1 + \sqrt{1 + (\omega CR)^2}\right]}{2u_{dc}R} \tag{18}$$

The theoretical waveforms in a fundamental period, including $i_{dc}$, $u_o$, and switching signals are presented in Figure 7. Based on the operation mode, switching signals, DC-link current optimal reference, and control strategy, the theoretical waveform is divided into eight stages:

(1) Stage 1 ($t_0 - t_1$): $0 < u_o < u_{dc}$, $L_{dc}$ is charged in energy-supplying mode I; $S_1$ remains on-state; $S_0$ is turned on in the zero vector period to prevent $i_{dc}$ from continuously increasing.

(2) Stage 2 ($t_1 - t_2$): $-u_{dc} < u_o < 0$, $L_{dc}$ is charged in energy-supplying mode II; $S_3$ remains on-state; $S_0$ is turned on in the zero vector period to prevent $i_{dc}$ from continuously increasing.

(3) Stage 3 ($t_2 - t_3$): $u_o < -u_{dc}$; $L_{dc}$ discharges in energy-supplying mode II due to $i_{dc}$ being still greater than $i_{dc}^*$; S0 is also turned on in the zero vector period; $i_{dc}$ begins to decrease.

(4) Stage 4 ($t_3 - t_4$): $u_o < -u_{dc}$ and $i_{dc} < i_{dc}^*$; to prevent $i_{dc}$ further decrease, S4 is turned on in the zero vector period, since $i_{dc}^*$ is the optimal reference of the DC-link current; $i_{dc}$ will be clamped near $i_{dc}^*$.

(5) Stage 5 ($t_4 - t_5$), Stage 6 ($t_5 - t_6$), Stage 7 ($t_6 - t_7$), and Stage 8 ($t_7 - t_8$) are similar to Stage 1 ($t_0 - t_1$), Stage 2 ($t_1 - t_2$), Stage 3 ($t_2 - t_3$), and Stage 4 ($t_3 - t_4$), respectively.

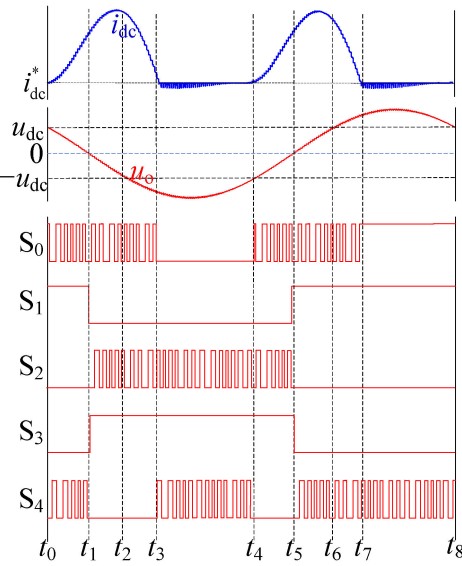

**Figure 7.** The theoretical waveforms in a fundamental period.

## 5. Control Strategy of the Output Voltage

Since the PI controller is not suitable for AC models, the math model of the singe phase CSI is established in d-q frame. The fundamental component of $u_\mathrm{o}$ is represented as follows:

$$u_\mathrm{o}(t) = u_\mathrm{od} \sin(\omega t) + u_\mathrm{oq} \cos(\omega t) \tag{19}$$

where $u_\mathrm{od} = U$, and $u_\mathrm{oq} = 0$. The quadrature virtual component of $u_\mathrm{o}$ is introduced, which is shown as follows:

$$u'_\mathrm{o}(t) = -U_\mathrm{o} \sin(\omega t - \frac{\pi}{2}) \tag{20}$$

The d-q frame components $u_\mathrm{od}$ and $u_\mathrm{oq}$ can be obtained as follows:

$$\begin{cases} u_\mathrm{od}(t) = u_\mathrm{o}(t) \sin(\omega t) + u'_\mathrm{o}(t) \cos(\omega t) \\ u_\mathrm{oq}(t) = u_\mathrm{o}(t) \cos(\omega t) - u'_\mathrm{o}(t) \sin(\omega t) \end{cases} \tag{21}$$

The same representation for $i_\mathrm{o}$ is shown as follows:

$$i_\mathrm{o}(t) = i_\mathrm{od} \sin(\omega t) + i_\mathrm{oq} \cos(\omega t) \tag{22}$$

where $i_\mathrm{od}$ and $i_\mathrm{oq}$ are the DC component. Substituting Equations (21) and (22) with (2), it yields the following:

$$\begin{cases} C\frac{du_\mathrm{od}}{dt} + \frac{u_\mathrm{od}}{R} = \omega C u_\mathrm{oq} + i_\mathrm{od} \\ C\frac{du_\mathrm{oq}}{dt} + \frac{u_\mathrm{oq}}{R} = -\omega C u_\mathrm{od} + i_\mathrm{oq} \end{cases} \tag{23}$$

$u_\mathrm{od}$ and $u_\mathrm{oq}$ are adjusted by the PI controller, and feedback decoupling is adopted to cancel out the coupling terms $\omega C u_\mathrm{od}$ and $\omega C u_\mathrm{oq}$. The control strategy of $u_\mathrm{o}$ is shown in Figure 8, where $u^*_\mathrm{od}$ and $u^*_\mathrm{oq}$ are the reference of $u_\mathrm{od}$ and $u_\mathrm{oq}$, and $i^*_\mathrm{od}$ and $i^*_\mathrm{oq}$ are the reference of $i_\mathrm{od}$ and $i_\mathrm{oq}$, respectively. $i^*_\mathrm{od}$ and $i^*_\mathrm{oq}$ are expressed as follows:

$$\begin{cases} i^*_\mathrm{od} = \frac{k_\mathrm{p}s + k_\mathrm{i}}{s}(u^*_\mathrm{od} - u_\mathrm{od}) - \omega C u_\mathrm{oq} \\ i^*_\mathrm{oq} = \frac{k_\mathrm{p}s + k_\mathrm{i}}{s}(u^*_\mathrm{oq} - u_\mathrm{oq}) + \omega C u_\mathrm{od} \end{cases} \tag{24}$$

where $k_\mathrm{p}$ and $k_\mathrm{i}$ are the proportional coefficient and integral coefficient, respectively.

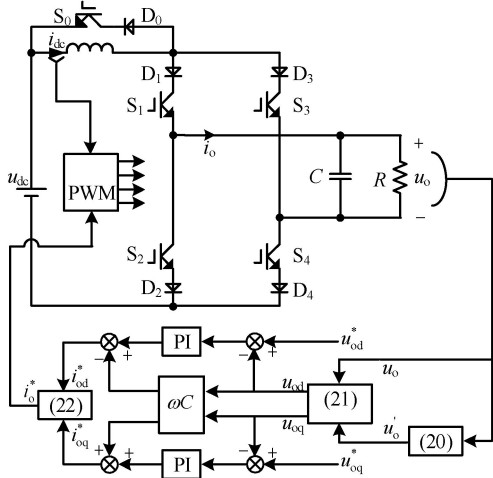

**Figure 8.** Control strategy diagram of output voltage.

Considering the digital delay, the open-loop transfer function $G_{\text{open}}(s)$ can be derived as follows:

$$G_{\text{open}}(s) = \frac{k_{\text{p}}s + k_{\text{i}}}{s(1 + T_{\text{s}}s)(1 + RCs)} \tag{25}$$

where $T_{\text{s}}$ is the switching period and is equal to 100 µs. Zero point is set to offset the pole, and the cutoff frequency is set at 1 kHz. $k_{\text{p}}$ and $k_{\text{i}}$ are set as $2000\pi RC$ and $2000\pi$. The Bode plot of $G_{\text{open}}(s)$ is shown in Figure 9a, and the Bode plot of the closed-loop transfer function $G_{\text{close}}(s)$ is shown in Figure 9b. Good track performance and fast dynamic response can thus be achieved.

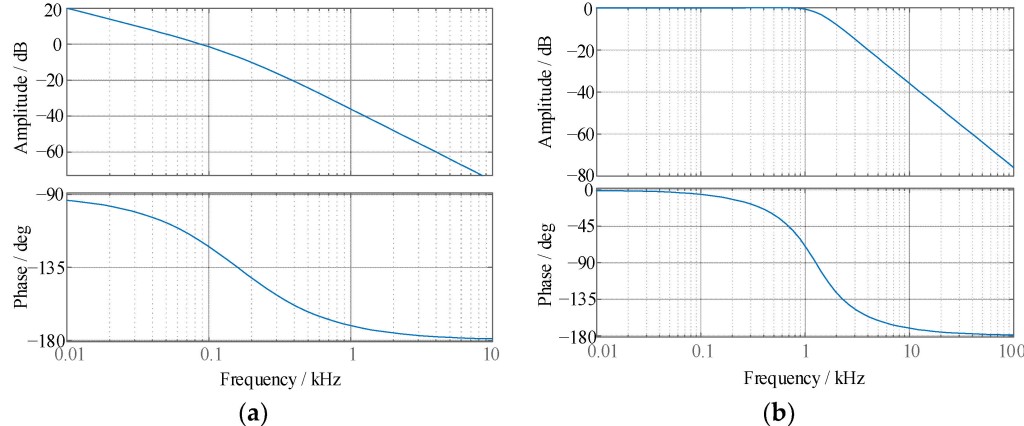

**Figure 9.** The Bode-plots of the output voltage control: (**a**) $G_{\text{open}}(s)$, (**b**) $G_{\text{close}}(s)$.

## 6. Experimental Results

An experimental prototype of a single-phase CSI is established and shown in Figure 10. The corresponding soft simulation is presented in the Supplementary Material. The algorithm of modulation and control are implemented by TMS320F28335, and the IGBT and diode are PM400HSA120 and RM300HA-24F, respectively. $u_{\text{dc}}$ is supplied by a DC power supply. The parameters of the passive components are consistent with Table 4.

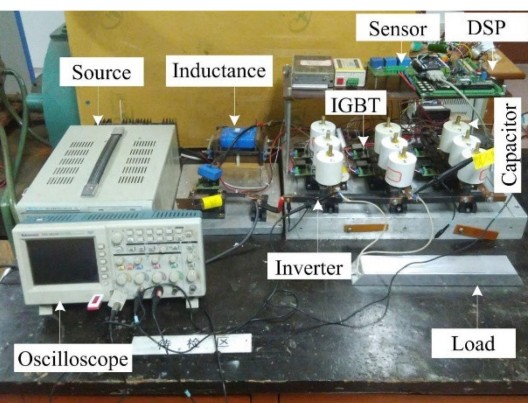

**Figure 10.** Experimental prototype of proposed single-phase CSI.

**Table 4.** Parameters of experiment.

| Name | Value |
|---|---|
| DC input voltage (V) | 25 |
| AC output voltage (V) | 30~150 |
| Load power (W) | $\leq 10^3$ |
| Output frequency (Hz) | $\leq 500$ Hz |
| Switching frequency (Hz) | 10 k |
| Sampling frequency (Hz) | 10 k |
| $L_{dc}$ (mH) | 4 |
| $C$ (μF) | 265 |
| $R$ (Ω) | 25 |

### 6.1. Experimental Results of Steady-State

The amplitude and frequency of the reference of $u_o$ are 50 V and 50 Hz. $i^*_{dc}$ is set to 13.5 A. The experimental waveforms of $i_{dc}$ and $u_o$ are shown in Figure 11a, where the amplitude and frequency of $u_o$ are consistent with the reference, and $i_{dc}$ is maintained in the range of 13.5 A to 15 A. Figure 11b shows that $u_o$'s THD is only 0.61%, and its harmonics are limited.

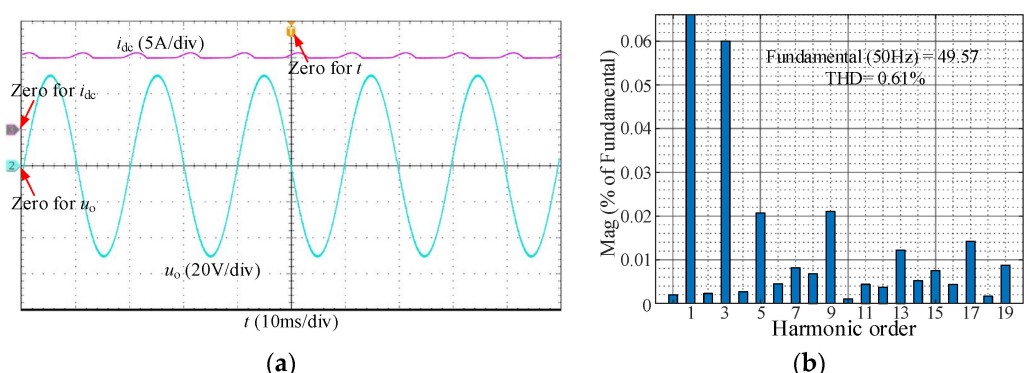

(**a**)            (**b**)

**Figure 11.** Experimental results under $i^*_{dc}$ = 13.5 A: (**a**) experimental waveforms of $i_{dc}$ and $u_o$; (**b**) FFT results.

In order to verify that $i^*_{dc}$ = 13.5 A is the optimal reference of the DC-link current, a steady-state experiment is carried out in Figure 12, where $i^*_{dc}$ is set to 11.5 A. A large fluctuation occurs in $i_{dc}$, which cannot maintain above 11.5 A. In some switching periods, $i_{dc}$ cannot meet the requirement of current for the AC load. Therefore, significant low-order harmonic distortion occurs in $u_o$, whose THD is 23.88%, and fundamental amplitude is 38.5 V, lower than the reference 50 V.

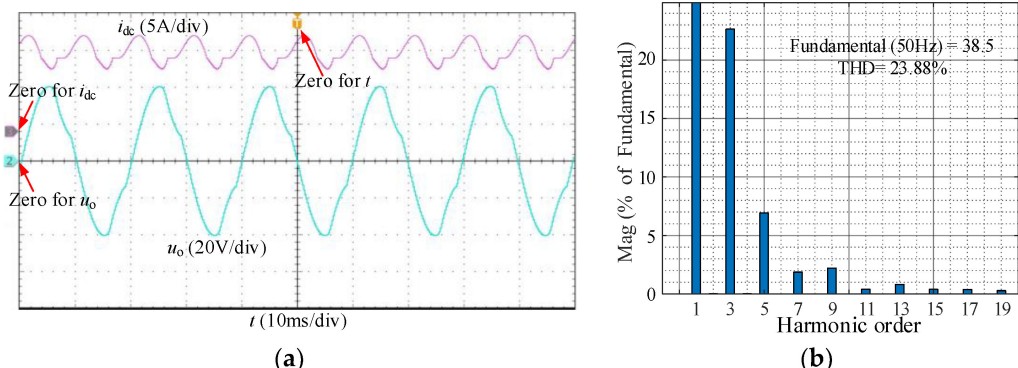

**Figure 12.** Experimental results under $i_{dc}^{*}$ = 11.5 A: (**a**) experimental waveforms of $i_{dc}$ and $u_o$; (**b**) FFT results.

In Figure 13, $i_{dc}^{*}$ is set to 20 A, and $i_{dc}$ maintains above 20 A with small fluctuation. Since $i_{dc}$ is higher than the amplitude of $i_o$, $u_o$ can track the reference. $u_o$'s THD is 1.38%, greater than the one when $i_{dc}^{*}$ = 13.5 A. Meanwhile, the switching loss and conduction loss of IGBT and diode are positively related to $i_{dc}$. The above experimental results show that the calculation method for the optimal reference of DC-link current is correct and feasible.

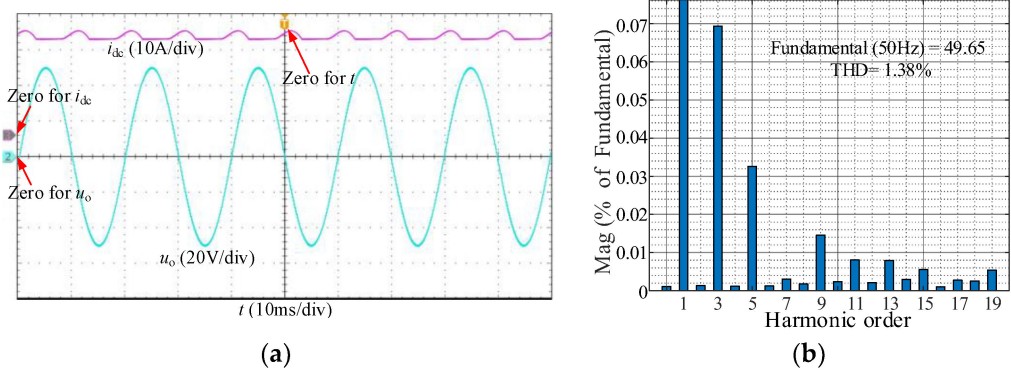

**Figure 13.** Experimental results under $i_{dc}^{*}$ = 20 A: (**a**) Experimental waveforms of $i_{dc}$ and $u_o$; (**b**) FFT results.

### 6.2. Experimental Results of Dynamic-State

The dynamic performance of the output voltage control strategy will be verified in this section. The frequency remains 50 Hz, and the amplitude of the reference voltage is adjusted from 40 V to 60 V. According to Formula (18), $i_{dc}^{*}$ is set to 9 A and 17.5 A, respectively. The experimental waveforms of $i_{dc}$ and $u_o$ are shown in Figure 14. $i_{dc}$ reaches the steady state again after 2 ms and remains above 17.5 A. In Figure 14, the $u_o$'s amplitude is adjusted to 60 V with smooth changing.

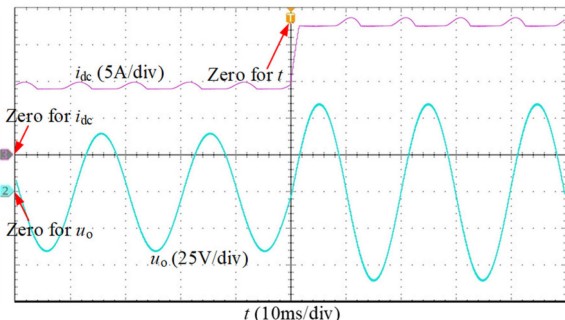

**Figure 14.** The dynamic-state experimental waveforms under amplitude changing.

Figure 15 shows the experimental waveforms when the amplitude is set to 50 V, and the frequency $\omega$ changes from 50 Hz to 100 Hz. According to (18), $i_{dc}^*$ is related to $\omega$, so it should be adjusted from 13.5 A to 16.5 A. In Figure 15, $i_{dc}$ and $u_o$ can track the reference quickly. In conclusion, the superior steady-state and dynamic-state performance of the output voltage control strategy can be fully proved, and the DC-link current reference from the proposed calculation method is the minimum value that can meet the power demand.

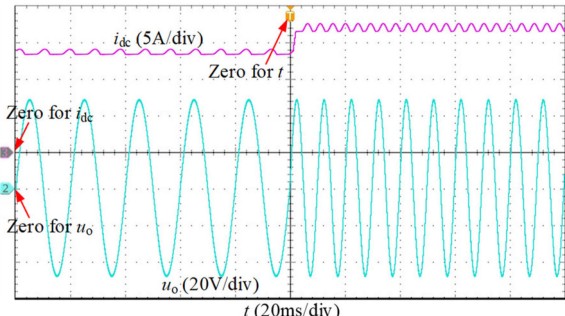

**Figure 15.** The dynamic-state experimental waveforms under frequency changing.

Finally, the CSI's efficiency has been tested under different load powers from 100 W to 500 W. A curve map of the CSI's efficiency is illustrated in Figure 16. When the load power increases, the power loss in the conduction and switching increases relatively more slowly than the load power, so the CSI's efficiency rises with the increased load power.

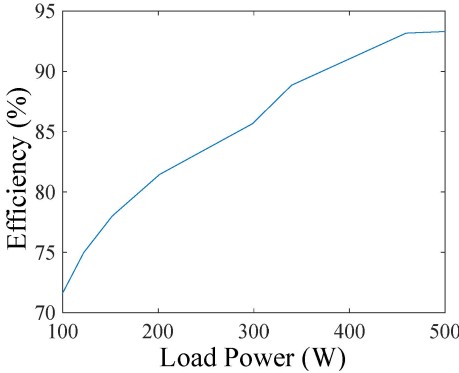

**Figure 16.** The efficiency map of the proposed single-phase CSI.

## 7. Conclusions

This paper proposes a novel PWM modulation method to control DC-link current for the improved topology of single-phase CSI; the corresponding operating modes and modulation strategy with DC-link current hysteresis control are also introduced in detail. The relationship between the optimal reference of DC-link current and output voltage is derived. A voltage control strategy based on d-q frame components is discussed. A series of simulations and experiments is set up to demonstrate the feasibility of the proposed method. The conclusions are summarized as follows:

(1) DC-link current can remain in the expected range by adopting the improved PWM modulation, and the number of switching activities is the same as the traditional modulation.

(2) The calculation method of optimal reference for DC-link current can meet the AC side load demand, improving current utilization and reducing loss.

(3) The control of DC-link current is implemented by switching magnetizing mode and freewheeling mode, which is separated from output voltage control. Thus, the DC-side model can be considered a controlled current source.

**Supplementary Materials:** The following supporting information can be downloaded at: https://www.mdpi.com/article/10.3390/en16186729/s1. Figure S1. Simulation waveforms in steady state: (a) DC-link current; (b) AC output volt-age. Figure S2. Simulation waveforms on the traditional single-phase CSI when the initial DC-link current is 0A: (a) DC-link current; (b) AC output voltage. Figure S3. Simulation waveforms on the traditional single-phase CSI when the initial DC-link current is 16A: (a) DC-link current; (b) AC output voltage.

**Author Contributions:** Methodology, Y.Z. and T.Y.; Software, T.Y.; Writing—original draft, Y.Z.; Writing—review and editing, Y.M. All authors have read and agreed to the published version of the manuscript.

**Funding:** This work is supported by the Youth project of science and technology research program of Chongqing Education Commission of China (No. KJQN202001105).

**Data Availability Statement:** The data presented in this study are available on request from the corresponding author. The data are not publicly available due to the funder's requirement.

**Conflicts of Interest:** The authors declare no conflict of interest.

## Nomenclature

| | |
|---|---|
| VSI | Voltage Source Inverter |
| CSI | Current Source Inverter |
| HVDC | High Voltage Direct Current |
| $L_{dc}$ | DC-link inductance (mH) |
| $C$ | AC side's filter capacitor ($\mu$F) |
| $R$ | AC side's resistive load ($\Omega$) |
| $i_{dc}$ | DC side's current (A) |
| $i_o$ | Output current on the resistive load $R$ (A) |
| $u_{dc}$ | DC side's voltage (V) |
| $u_o$ | Output voltage on the resistive load $R$ (A) |
| $p/q$ | Switching functions for $S_0$, $S_1$, $S_2$, $S_3$, and $S_4$ (-) |
| $d$ | Duty cycle (%) |
| $i_o^*$ | Reference of the output current $i_o$ (A) |
| $i_{dc}^*$ | Reference of the DC side's current $i_{dc}$ (A) |
| $\Delta i_{dc\_down}$ | Decrement of $i_{dc}$ when discharging (A) |
| $\Delta i_{dc\_up}$ | Increment of $i_{dc}$ when charging (A) |
| $T_s$ | Switching period (s) |
| $U$ | Fundamental amplitude of $u_o$ (V) |
| $I$ | Fundamental amplitude of $i_o$ (A) |
| $\theta$ | Initial phase of $i_o$ (rad) |
| $\omega$ | Fundamental frequency of $i_o$ (Hz) |
| $\Delta i_{dcmax}$ | Maximum reduction of $i_{dc}$ (A) |
| $u_{od}/u_{od}$ | d-q frame components of $u_o$ |
| $u_{od}^*/u_{oq}^*$ | Reference of $u_{od}/u_{od}$ |
| $i_{od}/i_{od}$ | d-q frame components of $i_o$ |
| $i_{od}^*/i_{oq}^*$ | Reference of $i_{od}/i_{od}$ |

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
