# Peer review of "Research on the Modulation and Control Strategy for a Novel Single-Phase Current Source Inverter"

_energies, doi:10.3390/en16186729_

Round 1

Reviewer 1 Report

This manuscript presents a novel single-phase current source inverter topology with the modulation, it is innovative in theory and application. However, some existing problems still need to be improved, the comments are as follows

Q1 English grammar needs improvement.

Q2 In introduction, several improved topologies of CSI are presented, please add a Table to provide a comprehensive comparison of several topologies.

Q3 Is Eq.(17) applicable for start up state?

Q4 Can this solution achieve the buck operation of CSI

English proficiency requires some modifications to improve the quality of the article

Author Response

  1. English grammar needs improvement.

The language of this paper has been carefully revised, and the grammatical errors and awkward expressions have been corrected.

  1. In introduction, several improved topologies of CSI are presented, please add a Table to provide a comprehensive comparison of several topologies.

The comparison among different CSI topologies are added in section 2, and shown in Table 3.

  1. Is Eq.(17) applicable for start up state?

Yes, Eq.(17) is also applicable for start-up state.

  1. Can this solution achieve the buck operation of CSI

This topology cannot work under the buck operation because the idc can only remain constant during the freewheeling model and cannot suppress the continuous decrease in DC-link current.

Reviewer 2 Report

The authors present a single-phase current source inverter topology. The work is supported by sound experimental evidence, but it still deserves further improvement before publication.

1) First, the literature review is poor, since it contains only 19 references. I would say that refs. [1]-[9] are unnecessary and were only added opportunely to increase the number of cited works. Only four CSI topologies are effectively analyzed in Section 1 in terms of [14]-[18]. The authors must perform an in-depth literature review and clearly identify both the research gaps and the contribution of their study.

2) The authors present the operating modes in Section 2, but the main theoretical waveforms that represent the circuit operation are missing. Please, consider including the ones related to the low- and high-frequency envelopes.

3) It seems that some variables remain undefined. Please, double-check the whole manuscript.

4) The mathematical analysis should comprise the current and voltage stresses on all semiconductors.

5) After presenting the proposed topology, it is necessary to compare it with similar approaches in terms of component count, number of drivers, stresses, and so on. A cost function (CF) could be used for this purpose. Please, check the expressions available in [R1] and [R2] for your reference.

[R1] Nascimento et al. (2022). Bidirectional Isolated Asymmetrical Multilevel Inverter. IEEE Transactions on Circuits and Systems II: Express Briefs, 70(1), 151-155.

[R2] Sun, et al. (2022). Seventeen-level inverter based on switched-capacitor and flying-capacitor-fed T-type unit. IEEE Access, 10, 33561-33570.

6) Section 5 is poor. An in-depth explanation of the control system must be provided, including the converter transfer functions and controller design.

7) Simulation results are redundant and MUST be removed because an experimental prototype is available.

8) The prototype specification MUST be summarized in a table including the dc input voltage, ac output voltage, load power, output frequency, and switching frequency, as well as any other relevant parameters.

9) It is necessary to present both a loss breakdown at the rated load condition and the efficiency plot as a function of the load power.

10) Proofread the manuscript, which contains some loose and unclear sentences.

Proofread the manuscript, which contains some loose and unclear sentences.

Author Response

  1. First, the literature review is poor, since it contains only 19 references. I would say that refs. [1]-[9] are unnecessary and were only added opportunely to increase the number of cited works. Only four CSI topologies are effectively analyzed in Section 1 in terms of [14]-[18]. The authors must perform an in-depth literature review and clearly identify both the research gaps and the contribution of their study.

The introduction has been revised to remove the redundant references. Meanwhile, some new literature, which is close to the research, has been added to the reference.

  1. The authors present the operating modes in Section 2, but the main theoretical waveforms that represent the circuit operation are missing. Please, consider including the ones related to the low- and high-frequency envelopes.

The theoretical waveforms in a fundamental period, including idc, uo, and switching signals, are presented in Figure 7. Based on the operation mode, switching signals, DC-link current optimal reference, and control strategy, the theoretical waveform is divided into eight stages:

1) Stage 1 (t0- t1): 0<uo<udc, Ldc is charged in energy-supplying mode I, S1 remains on-state, and S0 is turned on in the zero vector period to prevent idc from continuously increasing.

2) Stage 2 (t1- t2): -udc < uo <0, Ldc is charged in energy-supplying mode II, S3 remains on-state, and S0 is turned on in the zero vector period to prevent idc from continuously increasing.

3) Stage 3 (t2- t3): uo <-udc, Ldc discharges in energy-supplying mode II, due to idc is still greater than , S0 is also turned on in the zero vector period, idc begins to decrease.

4) Stage 4 (t3- t4): uo<-udc and idc <, to prevent idc further decrease, S4 is turned on in the zero vector period, sinceis the optimal reference of the DC-link current, idc will be clamped near.

5) Stage 5 (t4-t5), Stage 6 (t5-t6), Stage 7 (t6-t7), and Stage 8 (t7-t8) are similar to Stage 1 (t0-t1), Stage 2 (t1-t2), Stage 3 (t2-t3), and Stage 4 (t3-t4), respectively.

Figure 7. The theoretical waveforms in a fundamental period.

  1. It seems that some variables remain undefined. Please, double-check the whole manuscript.

A nomenclature table is added to explain the abbreviations and variables in this paper.

  1. The mathematical analysis should comprise the current and voltage stresses on all semiconductors.

Since the conduction current of all switching tubes is idc, the current stresses of all switching tubes are equal to the DC-link current. The voltage stresses can be derived according to Figure 2, and the voltage stresses of all switching tubes under different operating modes can be summarized in Table 2.

  1. After presenting the proposed topology, it is necessary to compare it with similar approaches in terms of component count, number of drivers, stresses, and so on. A cost function (CF) could be used for this purpose. Please, check the expressions available in [R1] and [R2] for your reference.

[R1] Nascimento et al. (2022). Bidirectional Isolated Asymmetrical Multilevel Inverter. IEEE Transactions on Circuits and Systems II: Express Briefs, 70(1), 151-155.

[R2] Sun, et al. (2022). Seventeen-level inverter based on switched-capacitor and flying-capacitor-fed T-type unit. IEEE Access, 10, 33561-33570.

Based on the definition in [12] and [13], a cost function (CF) is established in the following Equations to carry out a fair comparison among different CSI topologies.

where NV is the number of DC voltages, NUS is the number of unidirectional switches, NBS is the number of bidirectional switches, ND is the number of diodes, NC is the number of capacitors, Ndrv is the number of individual drivers, NT is the number of transformers, and Nl is the number of output current levels. The comparison among different CSI topologies in Figure X is presented in Table X. A new section (Section 2.2) is added to the paper for the comparison.

  1. Section 5 is poor. An in-depth explanation of the control system must be provided, including the converter transfer functions and controller design.

The converter transfer functions and controller design have been supplemented in section 5.

  1. Simulation results are redundant and MUST be removed because an experimental prototype is available.

The simulation result is probably redundant, but it can be a supporting material for the experiment. Thus, the section about the simulation result has been moved to Section S1 of the supplementary material of this paper.

  1. The prototype specification MUST be summarized in a table including the dc input voltage, ac output voltage, load power, output frequency, and switching frequency, as well as any other relevant parameters.

In the section about experimental results, a new table has been added, which describes the technical specification of the experimental prototype.

  1. It is necessary to present both a loss breakdown at the rated load condition and the efficiency plot as a function of the load power.

The CSI’s efficiency has been tested under different load powers from 100W to 500W. A curve map of the CSI’s efficiency is added to Figure 16. When the load power increases, the power loss on the conduction and switching increases relatively slower than the load power, so the CSI’s efficiency rises with the increased load power.

  1. Proofread the manuscript, which contains some loose and unclear sentences.

The language of this paper has been carefully revised, and the grammatical errors and awkward expressions have been corrected.

Reviewer 3 Report

Review of the article:

Research on the Modulation and Control Strategy for a Novel Single-phase Current Source Inverter

The authors present a new concept for the single-phase current source inverter. The main advantage of the proposed design is better control of the current, flowing through the input inductor. This is done by introducing an additional, fifth switch. The authors also present the inductor current control method using a hysteresis controller. The proposed inverter design concept was verified using simulation and later with practical laboratory experiments.

Reviewer comments:

-        The text in Figure 2 is a little too small. This should be increased. In addition, a short description of (a), (b), (c), and (d) should be included in the picture label,

-        What is the switching frequency of the simulated and tested inverter,

-        The negative oscillations in Figure 7a should be explained in the text.

-        What are switching losses compared to the conventional current source inverter? How does an additional switch impact the switching loss?

-        What are the main disadvantages of the proposed design?

Author Response

  1. The text in Figure 2 is a little too small. This should be increased. In addition, a short description of (a), (b), (c), and (d) should be included in the picture label,

Figure 2 has been revised, and the short descriptions for the sub-titles have been added.

  1. What is the switching frequency of the simulated and tested inverter,

The switching frequency is 10kHz, which is now listed in the newly added Table 1.

  1. The negative oscillations in Figure 7a should be explained in the text.

The theoretical waveforms in a fundamental period, including idc, uo, and switching signals, are presented in Figure 7.

The negative oscillations occur in stage 4 (t3 - t4), when uo<-udc and idc <. During the energy-supplying mode, Ldc discharges to the load. To prevent the further decrease of idc, S4 is turned on in the zero vector period, Ldc is charged by udc, and idc increases to . Sinceis the optimal reference of the DC-link current, the charging and discharging processes can achieve balance, and idc is clamped near.

  1. What are switching losses compared to the conventional current source inverter? How does an additional switch impact the switching loss?

In Figure 7, there are only two switching tubes in the activation state during any carrier period, which is the same as a conventional current source inverter. When idc <, S0 will replace the original switching tube for activation, so an additional switching only adjusts the DC-link current and will not increase the switching loss.

  1. What are the main disadvantages of the proposed design?

The proposed design can not work in the mode of buck and energy feedback, which is the main disadvantage.

Round 2

Reviewer 2 Report

The authors put significant effort into revising the manuscript according to my concerns. I have no more questions.